# Exploration of Nanosilver Calcium Alginate-Based Multifunctional Polymer Wafers for Wound Healing

**DOI:** 10.3390/pharmaceutics15020483

**Published:** 2023-02-01

**Authors:** Ernest Man, Claire Easdon, Iain McLellan, Humphrey H. P. Yiu, Clare Hoskins

**Affiliations:** 1Technology and Innovation Centre, Department of Pure and Applied Chemistry, University of Strathclyde, Glasgow G1 1RD, UK; 2School of Computing, Engineering & Physical Sciences University of the West of Scotland, Paisley PA1 2BE, UK; 3School of Engineering and Physical Sciences, Institute of Chemical Sciences, Herriot Watt University, Edinburgh EH14 4AS, UK

**Keywords:** nanosilver, cross-linked, alginate, wound healing

## Abstract

Wound care is an integral part of effective recovery. However, its associated financial burden on national health services globally is significant enough to warrant further research and development in this field. In this study, multifunctional polymer wafers were prepared, which provide antibacterial activity, high cell viability, high swelling capacity and a thermally stable medium which can be used to facilitate the delivery of therapeutic agents. The purpose of this polymer wafer is to facilitate wound healing, by creating nanosilver particles within the polymer matrix itself via a one-pot synthesis method. This study compares the use of two synthetic agents in tandem, detailing the effects on the morphology and size of nanosilver particles. Two synthetic methods with varying parameters were tested, with one method using silver nitrate, calcium chloride and sodium alginate, whilst the other included aloe vera gel as an extra component, which serves as another reductant for nanosilver synthesis. Both methods generated thermally stable alginate matrices with high degrees of swelling capacities (400–900%) coupled with interstitially formed nanosilver of varying shapes and sizes. These matrices exhibited controlled nanosilver release rates which were able to elicit antibacterial activity against MRSA, whilst maintaining an average cell viability value of above 90%. Based on the results of this study, the multifunctional polymer wafers that were created set the standard for future polymeric devices for wound healing. These polymer wafers can then be further modified to suit specific types of wounds, thereby allowing this multifunctional polymer wafer to be applied to different wounding scenarios.

## 1. Introduction

The national health service (NHS) of the United Kingdom spent in excess of £5.3 billion on wound care in 2017, but £3.2 billion of this sum was spent on wounds that did not fully heal [1]. It is quite clear that there is an incentive to develop an effective and financially sustainable wound-healing strategy that can offset these enormous costs. Effective wound-healing strategies follow the three key aspects of antibacterial activity, facilitation of increased cellular proliferation and the maintenance of a localised moist environment that supports wound regeneration [2]. Antibacterial activity and cellular proliferation can be expedited through the means of drug delivery, whilst the maintenance of localised environmental conditions relies on the use of a physical medium to help sustain and facilitate it. Despite the requirements for an effective wound-healing strategy, it is also important to consider the production costs associated with developing such a strategy. Undoubtedly, there are materials that can elicit superior therapeutic effects with respect to wound healing, e.g., epidermal growth factors [3] and collagen [4]. However, these are typically not financially viable for use for the large majority of the population, hence restricting their applicability to those who can afford it. In this regard, a multifunctional polymer device can be developed to facilitate all three aspects required for wound regeneration, whilst minimising the cost of production.

Given these specific criteria, various reagents have been tested to facilitate various aspects of wound regeneration, [2]. Nanosilver has been selected primarily for its antibacterial activity, as it provides an alternative form of antimicrobial treatments. This is highly important given the current state of antibiotic resistance in wounds [5], whereby 4.95 million deaths were associated with 88 pathogen-drug combinations in the year of 2019 [6]. Nanosilver is a well-explored field of research, especially in the aspect of antibacterial activity, where it induces antimicrobial activity through a variety of different mechanisms, such as the signal modulation of transduction pathways and facilitation of cellular toxicity through the build-up of reactive oxygen species (ROS), which leads to oxidative stress. Nanosilver also adheres onto cell walls and membranes which disrupts the permeability and stability of the membrane, as well as the particle penetration of the cell, leading to oxidative damage [7].

Alginate had been widely used as the physical medium for facilitating substance delivery and for the maintenance of a localised moist environment [8]. Hence, alginate has been well studied for regenerative medicine due to its biocompatibility, modifiability, and low cost [9]. On the other hand, aloe vera shows a myriad of regenerative properties, including the stimulation of fibroblast proliferation and migration, enhancement of wound contraction, reduction of tissue size, acceleration of reepithelisation, etc. [10]. These two specific reagents were also chosen for their use in fabricating nanosilver [11,12] from silver nitrate, which provides a greener alternative to standard methods of synthesis. In a standard nanosilver synthesis, reducing agents such as sodium citrate [13,14] and sodium borohydride [15] are used, followed by capping agents such as cetyltrimethylammonium bromide (CTAB) [16] and poly(vinylpyrrolidone) [17]. These additional reagents can be cytotoxic, resulting in irritation [18] and adding an additional layer of complication in materials production, such as a more rigorous wash regime to remove these cytotoxic reagents, ultimately increasing the production costs. In this regard, the use of both aloe and alginate can mitigate this problem as they are innately non-cytotoxic, thereby not requiring any stringent wash steps, which saves on production cost and time.

The primary rationale for choosing the combination of aloe, alginate and nanosilver is the possible synergy from their combined use. This would save a substantial amount of production time and cost as this would allow the nanosilver particles to be formed within the polymer. This study aims to discover possible synergies or antagonistic effects that may exist in the fabrication of wound-regenerative devices through two specific methods. This will give insight into the necessary compromises that may need to be undertaken if this synthetic method were to be employed for large-scaled production.

For the purposes of the study, two methods will be tested and compared. The first method, producing nanosilver alginate (NA), will follow the creation of silver nanoparticles through the use of alginate as the reductant and stabilising agent in a one-pot synthesis, whilst the other method, producing nanosilver aloe alginate (NAA), will explore the simultaneous use of aloe and alginate in varying concentrations, so as to determine the effects on the nanosilver produced. Samples from both methods will be tested for their cell viability, which will ascertain their potential feasibility for wound regenerative applications. For both methods, Ca^2+^ is used as a cross-linker to provide structural integrity to the alginate polymer, but also for the purpose of stimulating haemostasis in a wound-regeneration scenario [19].

## 2. Materials and Methods

### 2.1. Materials

HFF-1 Human foreskin fibroblasts were sourced from ATCC (Manassas, Virginia, VA, USA) trypan Blue Solution 0.4% (*w*/*v*) in PBS was sourced from Corning (Corning, New York, NY, USA), trypsin-EDTA (0.25%) phenol red, Dulbecco’s Modified Eagle Medium (DMEM), penicillin-Streptomycin (10,000 U/mL), Fetal Bovine Serum (FBS), L-Glutamine (200 mM), thiazolyl blue tetrazolium bromide, silver nitrate, calcium chloride pellets and sodium alginate powder were all sourced from Sigma-Aldrich (St. Louis, MO, USA). PBS Tablets were sourced from Fisher BioReagents, aloe vera gel was sourced from Holland and Barrett and Lysogeny broth (LB) (miller) was sourced from Merck (Darmstadt, Germany).

### 2.2. Preparation of Polymer Based Silver Nanoparticles

#### 2.2.1. Nanosilver Calcium Alginate Fabrication

A 0.1 mL solution of 0.1 mol/L silver nitrate was added to a solution of 10 mL of 1% weight: volume sodium alginate solution, which was then stirred continuously for 1.5 h. The solution (3 mL) was then poured into a small petri dish where it was then left in a LEC Medical Freezer LSFSF39UK (Prescot, Merseyside, UK) at −21 °C for 48 h. These samples were then freeze dried at 0.3 bar and −52 °C in a Christ Alpha 1-2 LD plus freeze dryer (Düsseldorf, Germany) for a duration of 72 h, before undergoing cross-linking in 10 mL 0.1 mol/L calcium chloride solution for 0.5 h per sample. After cross-linking, the alginate polymers underwent a triplicate wash with deionised water before subsequently undergoing the freeze-drying step again. This was then repeated for 1–10% weight: volume sodium alginate solution to generate 10 separate solutions.

#### 2.2.2. Nanosilver Aloe Calcium Alginate Fabrication

An aloe solution (1 L of 0.1 mol/L) was produced by dissolving 100 mL of pure aloe in 900 mL of deionised water. Sodium alginate powder (2.4 g) was dissolved in 240 mL 0.1 mol/L aloe solution for 24 h to produce a 1% *w/v* alginate: liquid aloe solution. Sodium alginate powder (12 g) was dissolved in 240 mL 0.1 mol/L aloe solution for 24 h to produce a 5% *w/v* alginate: liquid aloe solution. Sodium alginate powder (24 g) was dissolved in 240 mL 0.1 mol/L aloe solution for 24 h to produce a 10% *w/v* alginate: liquid aloe solution.

To each of the 240 mL alginate aloe solutions, 2.4 mL of 0.1 mol/L silver nitrate was added to create an alginate aloe silver solution which was stirred at room temperature under standard conditions for a total 24 h. A 30 mL aliquot of solution was extracted from the alginate aloe silver solution at the time points of 1 hr, 3 h, 5 h, 7 h, 9 h, 11 h and 24 h. From each of these 30 mL solution samples, 3 mL was ejected onto 10 separate mini petri dishes. These were then placed in a LEC Medical Freezer LSFSF39UK for 48 h before they were then moved to the Christ Alpha 1-2 LD plus freeze dryer for a period of 72 h under the conditions of 0.3 bar and −52 °C. After freeze drying, all of the samples were then submerged in a 10 mL 0.1 mol/L calcium chloride solution for 0.5 h before a triplicate wash with deionised water. This, in turn, created polymer wafer disks that were 2 mm, 3 mm and 4 mm in thickness with respect to 1%, 5% and 10% *w/v* alginate.

### 2.3. Characterisation Method

#### 2.3.1. ICP-OES

Samples of 3 cm diameter and 0.5 cm thickness were submerged in 40 mL deionised water for 24 h under standard conditions with stirring. Supernatant (10 mL) was extracted and filtered through a 1.2 μm syringe filter to remove any alginate fragments. Supernatant (1 mL) was then removed and combined with 1 mL aqua regia before undergoing ICP analysis to determine the quantity of nanosilver that had leeched from the polymer wafer. The samples were then analysed on a Perkin Elmer Avio500 (Waltham, MA, USA) and compared to and silver ICP standard calibration.

#### 2.3.2. SEM (Pore Size Measurement)

Fully desiccated polymer matrices were analysed via the secondary electron mode on a Quanta FEG Scanning electron microscope (Waltham, MA, USA) under high vacuum at 1.4 kV with a spot size of 4 at 10 kV. These were analysed using the secondary electron detector mode.

#### 2.3.3. SEM (Silver Size Measurement)

NA polymer samples were fully dissolved in 40 mL deionised water within a 50 mL Corning centrifuge tube. The samples were then centrifuged in a Thermo Scientific Heraeus Multifuge X1R centrifuge (Waltham, MA, USA) for 5 min at 14,500 rpm. The supernatant was discarded, and the centrifuge tube was then filled with 10 mL absolute ethanol, thereby submerging the pellet for 10 min. The ethanol was then discarded without disturbing the pellet. The pellet was then then gently dislodged from the centrifuge tube and transferred into a glass vial where it then air dried for 1 h. 

5 *w/v* and 10% *w/v* NAA polymer samples were fully dissolved in 40 mL deionised water within a 50 mL Corning centrifuge tube. The samples were then centrifuged in a Thermo Scientific Heraeus Multifuge X1R centrifuge for 20 min at 14,500 rpm. The supernatant was discarded without disturbing the hydrogel pellet and another 40 mL deionised water was added to the centrifuge before undergoing centrifugation using the same parameters. The supernatant was discarded, and the centrifuge tube was then filled with 10 mL absolute ethanol, submerging the pellet for 10 min. The ethanol was discarded, and the pellet was then transported into a glass vial where it was air dried for 1 h before analysis. 

Triplicate 1% *w/v* NAA polymer samples were fully dissolved in 40 mL deionised water within a 50 mL Corning centrifuge tube. The samples were then centrifuged in a Thermo Scientific Heraeus Multifuge X1R centrifuge for 0.5 h at 14,500 rpm. The supernatant was discarded without disturbing the hydrogel pellet, where 10 mL absolute ethanol was then added into the tube and left for 5 min. Excess aloe gel was then suspended within the absolute ethanol, which was then discarded. An additional 10 mL absolute ethanol was added into the tube, submerging the pellet, and was left for 20 min. The ethanol was discarded, and the pellet was then transported into a glass vial where it is air dried for 1 h before analysis. 

These samples were then analysed on a Quanta FEG Scanning electron microscope under high vacuum via secondary electron detector mode, with a spot size of 3 at 30 kV.

#### 2.3.4. FTIR Analysis

Fully desiccated polymer samples were analysed using a 0.4 cm^−1^ resolution Nicolet iS5 coupled with the iD5 ATR attachment (Thermofisher, Waltham, MA, USA). 16 scans were taken between 4000 to 400 cm^−1^ per sample utilising the ATR technique after background correction. 

#### 2.3.5. DSC and TGA Analysis

All samples were fully desiccated prior to analysis in a Christ Alpha 1-2 LD plus freeze dryer at −52 °C and 0.3 bar for 24 h. Samples weighing between 5 and 10 mg were analysed using a Q600 SDT thermobalance (TA Instruments, New Castle, U.K) for simultaneous DSC/TGA analysis. The samples were heated at a 5 °C/min heating rate from 20 °C to 250 °C, followed by a 30 min isothermal period before a 5 °C/min cooling from 250 °C to 20 °C. All experiments were carried out under purging nitrogen at a flow rate of 75 mL/min.

#### 2.3.6. Swelling Capacity and Evaporative Water Loss

Triplicate tests were conducted on all of the samples to ascertain the average swelling capacity. Samples were fully desiccated in a in a Christ Alpha 1-2 LD plus freeze dryer for 24 h at −52 °C and 0.3 bar pressure. The desiccated samples were weighed to determine their dry weight. These samples were then submerged in deionised water for a total of 24 h at room temperature and standard atmosphere before being patted down to remove excess water. The hydrated samples were then weighed to determine their maximum weight gain over the 24 h hydration period. The sample were then left to air dry at room temperature and standard atmosphere for 24 h, where weight measurements were taken at the time intervals of 2 h, 4 h, 6 h and 24 h to determine the evaporative water loss values.

#### 2.3.7. Bacterial Experiment

##### Bacterial Strain Preparation

*Escherichia coli* 25922 and methicillin-resistant *Staphylococcus aureus* BAA-1766 were obtained from ATCC, whereby all the bacterial strains were cultured in lysogeny broth (LB) (miller) Merck L3522 at 37 °C for 24 h in a MIR-H263-PE incubator. Optical density (OD_600_) was adjusted to 0.6, giving 4 × 10^8^ bacterial cells per mL.

##### Zone of Inhibition

*Staphylococcus aureus* and *Escherichia coli* inoculums were cultivated prior to the experiment and inoculums of approximately 2.11 × 10^8^ CFU/mL and 1.19 × 10^8^ CFU/mL were made up respectively [20], were then ejected and spread onto two separate lysogeny broth agar plates. Desiccated cross-linked polymer samples were hydrated in deionised water for 24 h before they were then cut out into 0.35 cm diameter disks using a circular punch. These disks were then freeze dried in a Christ Alpha 1-2 LD plus freeze dryer for 24 h at −52 °C under 0.3 bar pressure. After freeze drying, the sample discs were placed onto the inoculated agar plates in an equidistant manner prior to incubation in a Panasonic MIR-H263-PE incubator for 48 h at 37 °C. In conjunction with the NA and NAA samples, 0.35 cm diameter calcium alginate disks were also incubated with the bacterial cultures, so as to act as the control.

After the incubation period, the plates were then sprayed with a fine mist of 2.5 mg/mL 3-(4,5-dimethylthiazol-2-yl)-2,5-diphenyl-2H-tetrazolium bromide (MTT) to highlight areas of bacterial metabolic activity.

#### 2.3.8. Cell Culture Experiment

##### Cell Culture Preparation

Human foreskin fibroblasts (HFFs) obtained from ATCC were cultured in Dulbecco′s Modified Eagle′s Medium (Gibco) reinforced with 10% fetal bovine serum, 2% penicillin-streptomycin and 1% L-glutamine. These HFFs were then incubated in a PHCbi MCO-170AICUVH-PE incubator at 37 °C at 5% CO_2_ atmosphere until they reached a confluence of above 90%, at which point they were then split.

##### Cell Viability Assay

A single 0.35 cm sample disk was placed into the well of a 24-well plate, resulting in a total of 5 × 24-well plates containing triplicates of every sample between NA and NAA, with the addition of 3 cell-culture-only controls. Cell medium (500 µL) was placed into each well and then subsequently incubated for 48 h in a PHCbi MCO-170AICUVH-PE incubator (PHC, Tokyo, Japan) at 37 °C at 5% CO_2_ atmosphere. The medium contained within each plate was removed without disturbing the cells, and each well was then subsequently washed with 0.1 mL phosphate-buffered saline (PBS). trypsin-EDTA (0.1 mL, 0.05%) was added to each well in every plate and then incubated for 10 min at 37 °C in 5% CO_2_. Cell medum (0.1 mL) was ejected into each well, followed by the action of pipetting back and forth to evenly distribute the cells within the media. From each well, 20 µL was extracted and mixed with 20 µL 0.4% trypan blue solution. From this solution, 20 µL was extracted and ejected into an Invitrogen countess cell counting slide, which was then placed into an Invitrogen countess automated cell counter that counted the cell viability of the sample. The data obtained from this assay were then normalised against the control values, giving a relative cell viability percentage for each sample with respect to the control.

## 3. Results and Discussion

### 3.1. SEM (Polymer Matrix)

The results in Figure 1 depict the changes in pore size and topographical morphology with respect to increasing alginate *w/v* ratio for NA. The general trend from the pore size data, seen in Figure 1B, implies a very weak negative correlation with respect to pore size as a function of increasing alginate weight to volume ratio. The standard deviations for the pore sizes are generally quite large, ranging from ±11.6 to ±18.9 μm with outliers at 2% and 10% *w/v* ratio, which have a standard deviation of ±26.4 μm and ±5.7 μm, respectively. In regards to the topographical morphology the polymer samples, 1% and 2% *w/v* samples generally have a fused topography with pores that are harder to identify, whereas from 3–10% *w/v* samples, the pores and macrostructural alginate strands are more defined with visible undulations.

Based on the comparison of the NAA macrostructures, seen in Figure 2, it is quite clear that the higher ratio of alginate to aloe produces clearly defined pores with more consistent topographical features, as well as a flatter and more uniform morphology. Examples of this are the images in 1% *w/v* alginate, seen in Figure 2A, where high contrast areas are very prominent. The topography generally has large contrasts in regards to the heights between the features, which are further pronounced by the fusion of the alginate strands to a large unrefined mass. These areas are generally much darker due to the secondary electrons being obstructed, thus leading to poor visibility of said area. This occurs as a result of a large height difference between the area of interest and the surrounding topographical features, which blocks the passage of electrons, leading to a significant imbalance in lighting.

On the contrary, Figure 2C, 10% *w/v* alginate, has very defined features with uniform lighting throughout the entirety of the image, which imply that surface of these samples are significantly flatter with respect to 1% *w/v* alginate samples. In regard to the 5% *w/v* alginate samples, Figure 2B, there is a flatter morphology with the exception of the 1 h and 11 h samples, which have a deeper centre compared to the surrounding topography.

In terms of the comparison of these samples with respect to the changes in pore size against stirring time, Figure 2D, 1% *w*/*v,* has the lowest R^2^ value of 0.0122, which implies no association; however, 5% and 10% *w/v* have a R^2^ value of 0.556 and 0.4075, respectively, which implies a weak negative correlation. Considering the large fluctuations in standard deviation between the 1 h and 9 h in conjunction with the trend data, the general consensus would imply that there is no significant correlation between pore size and the stir time.

### 3.2. SEM (Silver Nanoparticles)

Based on the SEM analysis of NA nanoparticles seen in Figure 3A and Figure 4A, the particles appear to take upon a distinct cubic morphology at the lower % *w/v* values; however, as the % *w/v* increases, the nanoparticles lose their cubic morphology and slowly take on a more disorderly morphology, resembling a mixture of slightly disfigured rhombohedrons and octahedrons. As the % *w/v* values increase, the distances between vertices become less equidistant, leading to deformities relative to the cubic nanoparticle shapes. Based on the size analysis of NA particles in Figure 4A, there is a weak positive correlation between nanosilver size and increasing alginate concentration. The standard deviations do not follow any particular trend and vary greatly, with the smallest standard deviation being observed in 2% *w/v* ±22 nm and the largest being at 7% *w/v* ±118 nm. With respect to the average particle size, the standard deviations are relatively small given the large size ranges between average particle size. The average nanosilver particle sizes from this set range between 96 nm as the smallest and 834 nm as the largest. In terms of their size implications, the particles themselves are too big to penetrate through damaged skin [21] and so may be highly applicable to cutaneous wound healing without the risk of systemic accumulation.

Figure 3B displays the nanosilver particles generated from 1% *w/v* NAA, where the morphology of the particles takes on a spherical shape for all the samples within the stirring time of 1–24 h. Based on the size analysis in Figure 4B, there is a strong negative correlation between particle size and stirring time, where the average particle size ranges from 105.6 nm to 45.82 nm with respect to 3 h and 24 h stirring time.

In terms of the changes in standard deviation with respect to stir time, in Appendix A there is weak-medium negative correlation, where the smallest standard deviation is ±7.5 nm occurring at 9 h stir time, whilst the biggest is ±16.4 nm occurring at 7 h stir time.

Nanosilver particles created by 5% *w/v* NAA, Figure 3C and Figure 4C, demonstrate a morphology that is generally cubic in nature within the stirring times of 1–24 h. The cubic shape can easily be identified; however, the edges and vertices are not necessarily equidistant within each nano-cube, with morphological inconsistencies being present within each sample. In regards to the changes in particle size with respect to stirring time, there is no significant trend given the low R^2^ value of 0.1112. The largest average particle sizes belong to 1 h and 24 h stirring time, giving a size of 177.3 nm and 178.8 nm, respectively. In terms of the standard deviation for this sample set, the smallest occurs at 3 h, giving a range of ±22 nm, whereas the largest occurs at 24 h, giving a range of ±38 nm. In regard to the changes in particle size standard deviation with respect to stir time, in Appendix A, there is a medium positive correlation between the increase in particle size standard deviation and stir time. 

Particles created via 10% *w/v* NAA, Figure 3D and Figure 4D, demonstrate a mixture of particle morphologies, where they are primarily cubic in shape, but not necessarily with well-defined edges, as some particles have a slightly rounded edge. Based on observations, it appears that there is a trend whereby the increase in stirring time shifts the morphology from a rounded cubic shape to a more refined cubic shape with sharper and straighter edges. In regards to the changes in particle size, there is a weak positive correlation between stir time and particle size, with the smallest average size being 121 nm at 1 h stirring and the largest being 257 nm at 11 h stirring. The standard deviations for these samples generally fluctuate and are relatively large with respect to the average particle size, where the smallest set of standard deviations occur at 5 h stirring giving a deviation of ±15 m, whilst the largest occurs at 24 h stirring which is ±42 nm. In terms of the changes in particle size standard deviation relative to stir time, Appendix A, there is a weak-medium positive correlation, suggesting an increase in particle size standard deviation as stirring time increases. 

Based on the comparison between the nanoparticles produced by 1%, 5% and 10% *w/v* NAA, there appears to be a distinction in terms of the morphology of each sample set. 1% *w/v* are all spherically shaped, 5% *w/v* are distinctly cubic, whilst 10% *w/v* span between cubic to rounded cubic. 1–10% NA samples vary between cubic, rhombohedron and octahedron-like shapes. It may be implied that that relative concentration of aloe against alginate affected the nanoparticle morphology, whereby higher ratios of aloe to alginate result in spherical nanoparticles, whilst lower ratios of aloe to alginate result in morphologies with defined vertices, such as cubes, rhombohedrons and octahedrons. This is supported by studies that only utilise aloe and silver nitrate for nanosilver formation, whereby the particle morphology of the nanosilver is spherical [22,23]. Taking into account the fact that the NAA samples 1%, 5% and 10% *w/v* contained 1:1, 5:1 and 10:1 ratios of aloe:alginate, respectively, it may be implied that the predominant reducing agent in 1% *w/v* NA is aloe, given the spherically shaped nanosilver particles. Beyond the 1% *w/v* NAA, it is unknown whether a specific alginate concentration threshold results in a predominantly alginate-driven reduction process, so as to overshadow the aloe-driven reduction mechanisms. Given the fact that the particles generated from 10% *w/v* NA and 10% *w/v* 1–24 h NAA are morphologically different from one another it may be assumed that the alginate and aloe work in tandem to simultaneously reduce the silver, resulting in a unique morphology that differs from aloe or alginate only reduction processes. It can be assumed that the two reduction processes do not occur separately from one another, as it would therefore result in a mixture of relatively small spherical particles and relatively large cubic/rhombic particles, which are not present. 

In terms of the size comparison between all the samples, there appears to be trend in regards to the changes in particle size relative to the differing aloe:alginate ratios. Disregarding the correlations between particle size and stir time, 1% *w/v* NAA samples had an average particle size of 78.1 nm, 5% *w/v* NAA samples had an average of 153.4 nm, 10% *w/v* NAA samples had an average of 167.8 nm and NA samples overall had an average of 515.3 nm. This increase in average particle size with respect to decreasing aloe concentrations may suggest that the presence of aloe can limit the size of nanosilver particles that are undergoing reduction in alginate. 

Overall, in the context of wound regeneration, the NAA and NA samples are generally feasible options, as the particle sizes are above 21–45 nm, which defines the range in which nanoparticles will penetrate through damaged skin [21]. The only exception would be 1% *w/v* NAA 24 h stir time samples, which created particles with an average size of 45.8 nm ± 7.9 nm, possibly resulting in the penetration of damaged skin. This is an important consideration as nanosilver can accumulate within the host, leading to localised cytotoxicity within certain organs such as the lungs, kidney, spleen, and brain, etc. [24]. The general negative impacts of nanosilver on mammalian cells include genotoxic effects, cytotoxic effects and anti-proliferative effects, which all have the potential to destabilise the cell genome [25]. Due to this, it is important that the correct nanosilver sizes are chosen, so as to prevent host injury.

### 3.3. ICP-OES

The ICP data indicate that both fabrication methods release differing quantities of silver into the surrounding deionised water during the course of the 24 h submersion period. Based on the data obtained for the NA samples, in Figure 5A, 1–10% *w*/*v,* there are no discernible correlations in regards to the increase in silver release with respect to increasing weight to volume ratio. If the sample data for 1% *w/v* were to be considered outliers, then the range of silver release would be within 0.056 ppm and 0.074 ppm.

For the NAA samples, there were no discernible trends in regards to the increase in silver release with stirring time. In terms of the range of silver release values, 1% *w/v* had a range of 0.023–0.044 ppm, 5% *w/v* had a range of 0.024–0.042 ppm and 10% *w/v* had a range of 0.022–0.048 ppm. Comparison between Figure 5A,B suggests that the presence of aloe, within the confines of the alginate matrix, can help to diminish the release of silver into the surrounding environment. Given the fact that each sample was made up of 3 mL extractions, whereby each extraction originated from a bulk solution containing identical quantities of silver nitrate, each sample therefore contained 56,961.7 ppm worth of silver in each alginate matrix. From these samples, less than 0.0003% of the total silver content was released within the span of 24 h.

In regards to the possible effects of nanoparticle shape and size on the ICP release values, there generally appears to be no identifiable correlation. Given the randomised nature of ICP release values from the NAA samples, it can be implied that the smaller-sized nanosilver spheres are not released more readily than that of larger cuboidal nanosilver. Taking all this into account, coupled with the total release of particles relative to the theoretical amount of nanosilver formed within the sample, it may be implied that the nanoparticles are formed interstitially within the alginate matrix. The resultant nano-morphologies form in between the alginate strands, whereby the application of freeze-drying followed by cross-linking agent further constrains the nanosilver, resulting it being tightly bound within the cross-linked alginate strands, thus leading to a lower release value. 

The implications of this are quite important, as it affects the efficacy of nanosilver both in terms of its maximal antibacterial effects and cytotoxicity towards mammalian cells. The release rates may be relevant for drug release applications, where nanosilver can be utilised as a drug carrier for therapeutic purposes [26]. Assuming that it may be feasible to generate drug-loaded nanosilver within an alginate matrix, whilst being able to control the release rates of nanosilver, this could therefore lead to an extendable therapeutic window for sustained drug delivery.

### 3.4. FTIR Analysis

Given the fact that the primary material is alginate, all the samples shown in Figure 6 have a similar chemical profile to sodium alginate, Appendix A. The peak at 3000–3600 cm^−1^ represents OH stretching, the 2850–2980 cm^−1^ represents the cyclic C-H stretching, 1500–1700 cm^−1^ represents the asymmetric COO stretch and 1350–1500 cm^−1^ represents the COO symmetric stretch. 1250–1350 cm^−1^ represents the OCH-CCH stretch, 1050–1100 cm^−1^ represents the cyclic OCO stretch, 950–1150 cm^−1^ represents the CO stretch and 920–980 cm^−1^ represents CO stretch specific to uronic acids. 

For all of the samples, the positions of the peaks are essentially near identical to those of pure sodium alginate, Appendix A. In terms of the NA samples, the peak positions remained consistent between all 10 samples, indicating that the presence of nanosilver did not affect the affect the characteristics of the alginate matrix. Comparison of both sodium alginate [27,28] and aloe [29,30] spectra with existing literature indicate that both raw materials used are near identical to those found in other literature, thereby allowing for standardisation and repeatability when applying both alginate and aloe for nanosilver fabrication.

Comparison between the NA samples, Figure 6A and the NAA samples, Figure 6B–D, does not indicate any distinct differences in terms of chemical shift patterns; however, this may be due to the fact that the composition of aloe is relatively small compared to alginate. Each 1% *w/v* sample contains 0.03 g alginate per 0.1 mol/L 0.1 mL aloe solution, 5% *w/v* samples contain 0.12 g alginate per 0.1 mL 0.1 mol/L aloe solution and 10% *w/v* samples contain 0.3 g alginate per 0.1 mL 0.1 mol/L aloe solution. The relative compositions imply that the effects of aloe on the chemical shift pattern would be greatest in 1% *w*/*v*; however, no peaks specific to aloe were observed. Based on Appendix A, all of the key chemical shifts present in sodium alginate are present in aloe, with some stretches being slightly shifted, i.e., 1350–1500 cm^−1^ COO symmetric stretch and 1250–1350 cm^−1^ OCH-CCH stretch shifted upwards relative to sodium alginate. The most important differentiator between aloe and alginate lies in the presence of the 1743 cm^−1^ peak, which represents the C=0 stretching vibrational mode in the -COOCH_3_ carboxylic ester group of pectin [29]. Given the fact that the 1743 cm^−1^ OH stretch is not present within any of the NAA samples, it could be implied that the effects of aloe on the compositional nature of the polymer are negligible. 

### 3.5. Swelling Capacity and Evaporative Water Loss 

The swelling capacity defines a polymer’s ability to absorb and retain water within its matrix. Based on the data observed from Figure 7A NA samples, there is a moderate positive correlation in regards to the increase in swelling capacity with respect to the increase in alginate weight to volume ratio. Within this data set, 5% *w/v* provided the lowest average swelling capacity of 492%, whilst 10% *w/v* provided the highest averages swelling capacity of 752%. 

In terms of the data from the Figure 7B NAA samples, there is no discernible trend in regards to the stirring time on swelling capacity. This implies that the effects of nanosilver formed within the matrix minimally impacts the physiochemical parameters associated with water absorption. Within this data set, the 10% *w/v* samples provided the highest range of swelling capacity values with the lowest average value occurring at 7 h stir time, 724%, whilst the highest average occurred at 11 h stir time, 835%. 5% *w/v* samples had the second highest range of swelling capacity values, with the lowest average occurring at 1 h stir time, 567%, whilst the highest average occurs at 7 h stir time, 678%. 1% *w/v* samples had the lowest overall swelling capacity values, with the lowest occurring at 7 h stir time, 401%, whilst the highest values occur 11 h, 476%.

Comparison between the different fabrication methods suggests that swelling capacity is not only unaffected by the presence of nanosilver, but is also unaffected by the presence of aloe that is integrated into the polymer matrix. Due to the nature of the fabrication process, different factors such as pore size and matrix morphology can affect the surface area and volume of the polymer matrix, thereby causing fluctuations in water absorption and retention. This is reflected in Appendix A, which indicates that there are no significant correlations associated with the increase in standard deviation as a function of increasing alginate *w/v* and increasing stirring time.

Examination of the data from Figure 8A indicates a medium negative correlation between relative weight loss and increasing alginate weight to volume ratio. It should be noted the fluctuations in standard deviation are most prominent after the 8% *w/v* mark, where 8% and 9% *w/v* have a range of approximately ±15% weight loss, whilst 10% *w/v* has approximately ±10% standard deviation. 

The samples with a *w/v* ratio of less than 8% all have standard deviations that are substantially smaller, with a range between 0.74–3.7%. Given the fact that the samples with a lower *w/v* ratio generally retained less water, it could then be implied that that the lower standard deviation of evaporative water loss is as a result of the vast majority of water being lost, leaving only water that is deeply bound in the matrix. This would therefore explain why 8–10% *w/v* samples had a large range of standard deviations, as the rate of evaporation loss was enough to remove all the water contained within the pores of the matrix. It should also be noted that differences in the topography and morphology of the matrix can affect the total surface area, which in turn can affect the rate of evaporative water loss. Based on the data from NAA, Figure 8B, the relative evaporative weight loss of 1% *w/v* and 5% *w/v* is quite similar, falling within the range of 82–86%, coupled with standard deviation values between 1 and 4%. For 10% *w*/*v,* the average values fluctuate between 71 and 75% relative weight loss, but the standard deviation is ±5 to ±9%, which is substantially larger than those of the fluctuations in 1% and 5% *w*/*v*. The general implication is that the presence of aloe and nanosilver within the alginate matrix imparts minimal effect on evaporative water loss.

Overall, both NA and NAA samples provided substantial swelling capacity values, which is highly important in regards to its application in wound healing, specifically in the aspect of exudate absorption [31]. 

### 3.6. DSC and TGA

The thermal properties of the samples were studied using a combined DSC/TGA analysis. Based on the DSC and TGA data from Figure 9, all the samples were thermally stable up to approximately 200 °C, with no observable differences between different silver or aloe content. This implies that nanosilver and aloe did not affect the thermal properties of the alginate polymer. From the DSC analysis of NA samples Figure 9A, an endothermic event occurs between 224 and 232 °C, which coincides with the sudden weight loss event for these samples. This is likely due to thermally induced structural decomposition, such as the dehydration of -OH groups in alginate. Such decomposition events are observed in all the other samples; however, the range of occurrences differ between each sample type. 

For all NAA samples (1% *w*/*v*, 5% and 10% *w*/*v*), as shown in Figure 9B–D, slightly higher thermal stability was shown (up to around 210–218 °C) in that the endothermic dip is broader and lower in magnitude compared to that observed from the NA samples. This result suggested that aloe possibly reduces the intensity of the endothermic event of alginate, but further investigation would be required. Comparing these DSC data with those of pure aloe and sodium alginate, Appendix A, NA closely resembles that of pure sodium alginate. In contrast, none of the samples bear any resemblance to the DSC profile of aloe, suggesting that the NAA samples were not simple physical mixtures of aloe and alginate.

Regarding the TGA analysis, Figure 9E–H, all samples retain a similar profile, with the major weight change event occurring just beyond 210 °C. When comparing the TGA profiles of all the samples with pure aloe and sodium alginate, Appendix A, all the samples resemble that of sodium alginate, with a single weight change event, whereas aloe has two weight loss events. In this regard it can be implied that the effect of aloe on the thermal properties of nanosilver alginate is generally quite low. 

### 3.7. Zone of Inhibition

Based on the results presented in Figure 10 and Figure 11, it can be implied that the zone of inhibition is generally larger in NAA samples compared to NA samples.

Figure 11A depicts 1% *w/v* ratio having a relative zone size range between 1.38 and 2.02 mm radius, 5% *w/v* ratio ranges between 0.35 and 1.08 mm and 10% *w/v* ratio ranges between 0.25 and 1.35 mm, where the relative zone size is the difference between the zone of inhibition size and the sample disc size. From this specific data set, we can conclude that there are no overall identifiable trends relating the zone of inhibition size to increasing stir time. This also applies to NA samples, Figure 11B, where the zone of inhibition range is between 0.12–0.95 mm, which is coupled with an extremely low R^2^ value, implying no that there are no significant correlations relating the zone of inhibition size with the increase in alginate weight/volume ratio.

Taking into account the data presented by the nanoparticle SEM micrographs in Figure 3, it can be suggested that the relative zone size ranges are as a result of the nanoparticle size and shape. From the NAA samples, 1% *w/v* has the largest zone of inhibition of 1.38–2.02 mm radius, which is associated with spherical nanoparticles that have a size ranging between 45.8 nm to 105 nm. 5% *w/v* and 10% *w/v* NAA samples have a smaller zone of inhibition radii of 0.35–1.08 mm and 0.25–1.35 mm, respectively. For these two samples, the nanoparticle morphology is generally cubic, with their size ranging between 125.7 and 178.8 nm and 121.2 and 257.1 nm, with respect to 5% *w/v* and 10% *w*/*v*. This is also present in NA samples, which have a morphological mixture of cubes rhombohedrons and octahedrons, ranging from 96.1–985.5 nm, which is associated with the zone size range of 0.12–0.95 mm. This association can also be further linked to the ICP data from Figure 4, which implies that NAA samples all have roughly the same release values as one another but are still generally less than that of the NA samples. Overall, it appears that the spherical nanosilvers are able to elicit stronger antibacterial effects than their cuboidal counterparts; however, this may be due to their smaller average size, which may allow for improved penetration, as well as an increase in contact area, leading to improved antibacterial effect [32]. 

In terms of the data for the *E. coli* zone of inhibition experiments, the general consensus suggests that the concentration of nanosilver released may have been too low to inhibit the growth of *E. coli*, or that the size and morphology of the particles were simply ineffective against the bacterium. 

This is supported by the fact that nanosilver has been shown to elicit antibacterial effect on *E. coli* [33,34,35], where the primary mechanisms include the use of reactive oxygen species (ROS) generated from the nanosilver, coupled with the increase in NAD^+^ to NADH ratio within the bacteria which leads to internal ROS production [36]. Generally speaking, the zone of inhibition was not present for the *E. coli* samples and showed signs of the bacteria invading the sample discs, as shown by the accumulation of MTT dye in some of the samples, thereby implying a lack of effective antibacterial activity. Based on the results, it could be implied that MRSA is generally more sensitive to nanosilver compared to *E. coli* with respect to their relative response. Given the fact that the polymer samples were desiccated prior to their placement on the agar surface, they could therefore be hydrated prior to placement, so as to pre-establish the interface needed to facilitate nanosilver release, which in turn my increase the release rate over the 24 h period. 

### 3.8. Cell Viability

In terms of the relative average cell viability results for both NA and NAA samples, Figure 12, there appears to be a large fluctuation in values, with the lowest averaging around the 70% viability mark, whilst the highest averages around 120% viability. For 10% *w/v* NAA samples, the typical average is around 98.4%, whilst 5% *w/v* and 1% *w/v* have a typical average of 91.7% and 91.6% average respectively, Figure 12A. The highest viability values for 10% *w*/*v*, 5% *w/v* and 1% *w/v* are 120.9%, 107.1% and 122.5%, respectively, whilst the lowest values are 67.6%, 70.88% and 84%, respectively.

In terms of the standard deviation of the NAA samples, 10% *w/v* NAA generally has the lowest standard deviations with an average of ±9.4% and is quite concise with its range between ±5.4% and ±16.2%. 1% *w/v* NAA has an average standard deviation of ±13.4% and is the most concise with a range of ±9.7 and ±20.3%. 5% *w/v* NAA has the highest average of ±16.2% with the largest range of ±9.7% to ±26.9%. For NA samples, Figure 12B, the values average around 95.8%, with the largest and smallest averages being 125.8% and 69.8%, respectively. The range of standard deviations is quite large for this sample set, with 7% *w/v* being ±33.7%, whilst the smallest is observed at 10% *w*/*v*, ±7.9%. Based on the data from Figure 12A,B, there does not appear to be any significant trends between relative cell viability and the parameters of stir time and alginate weight to volume. 

Comparison of cell viability with respect to nanosilver can be observed in Figure 13A,B, which generally indicates that the 1% *w/v* NAA samples and the 10% *w/v* NAA samples have a higher cell viability than 5% *w/v* NAA samples, whilst NA samples are on average above 80% viability. 

Taking into consideration the release rates, in conjunction with the nanosilver particle size and morphology, it appears that the cytotoxic effects of nanosilver have been mitigated to an extent. This may be due to the fact that the nanosilver was made interstitially between the alginate molecules, thereby reducing the HFF’s exposure to the particles. In a standard nanosilver loading experiment, the silver is loaded into the medium via diffusion, thus saturating the carrier. When the carrier is then placed into the cell medium, the initial release of nanosilver may be acute, leading to cell death as it is up taken by the cell culture [37]. In the case of NAA and NA samples, it may be suggested that the release rates of nanosilver are below the acute toxicity level, thereby implying that the intracellular accumulation of ROS is below the threshold that causes oxidative stress and thus cell death. 

It should be noted that the viability results of the trypan blue method are affected by the fact that the calcium alginate polymer breaks down as it is incubated alongside the cell culture. It has been shown that fibroblasts can utilise external calcium sources to increase their proliferative capabilities [38,39,40]; however, in this instance, the calcium that is utilised is sourced from within the alginate polymer where it acts as the cross-linker holding the matrix together. As the calcium ions are consumed by the fibroblasts, the alginate begins to break down, leading to the formation of small calcium alginate fragments. Despite the PBS cleaning step, some of these smaller alginate fragments remain within the well, which then absorbs the trypan blue. Given the fact that these calcium alginate fragments can be as small as the trypsinised fibroblast cells, this can then result in a large number of false negatives during the cell counting process, as the automated cell counter identifies these alginate fragments as dead cells, leading to a cell viability percentage that is skewed downwards, away from the true value.

Amongst the cell viability percentages present within Figure 12, there are some values above the 100% normalised cell viability threshold, which can be explained by the polymer acting as Ca^2+^ source, resulting in increased HFF proliferation, whereas the control is lacking one. Based on the light microscopy images in Figure 14, we can clearly see that the presence of the alginate polymer affects the migration and proliferation of HFF cells. Figure 14A shows the control well without any polymers, where the HFF cell growth patterns are generally of a slightly lower density and well spaced out. On the contrary, Figure 14B–D are examples of wells containing the polymer nanosilver samples, where the red ring indicates the location of where the HFF cells were bound to the alginate polymer, prior to the polymer’s detachment. Figure 14B,D, gives an example of HFF growth patterns near the peripheral edges of the alginate molecule, whereas Figure 14C demonstrates the growth pattern of HFF cells where they are bound to the face of the polymer. Based on the observations, it is quite clear that the cell density increases as it localises towards the polymer, acting as an anchor point, thereby holding the polymer to the plate. This cellular localisation, increased proliferation and migration may be explained by the fact that the alginate matrix not only acts as another interface for cellular adhesion in the *Z* axis, but also Ca^2+^ for cell growth. In the control well, the proliferation and migration of cells are limited to the *X* and *Y* axis, resulting in limited growth as the two-dimensional surface area becomes more crowded with cells. The introduction of the alginate matrix adds an extra Z dimension which gives more directional growth potential for the HFF cells. The images in Figure 14B–D show that once the cells are seeded, the populations growing on the surface of the well connect with the ones on the polymer, leading to increased cell growth as the HFF can now grow upwards towards the polymer via the fibroblast bridge. Given the fact that the polymer is suspended within the cell media and is therefore free to float around, the minute vibrations of the incubator may be enough to cause the polymer to undulate slightly and therefore provide a subtle mechanical stimulus for increased fibroblast growth [41]. 

Aloe was included as a regenerative agent for enhancing fibroblast proliferation; however, the data comparison between Figure 12A,B, suggests that the effects of aloe were negligible. There are three possible reasons for this. The first reason is that the concentration of aloe is too low to elicit an effect, the second reason may be that the aloe is too deeply incorporated into the alginate matrix’s architecture, thereby preventing it from being released and the third reason may be due to the fact that the proliferative effects of Ca^2+^ simply overshadowed that of aloe. 

Overall, it can be assumed that the presence of a calcium alginate matrix is able elicit a multitude of advantages in regard to increased fibroblast growth, which in turn leads to a higher turnover of live cells relative to dead cells and thus a high degree of cell viability. These advantages include, Ca^2+^ release, three-dimensional cellular migration, and mechanical stimulation, all of which appear to be able to mitigate the cytotoxic effects of nanosilver particles of varying sizes and morphologies. 

## 4. Conclusions

In this study, we have demonstrated two simple methods of alginate-facilitated nanosilver synthesis to form a multifunctional polymer wafer for wound healing. Both methods generate samples that can elicit controlled nanosilver release, which display antibacterial effects towards MRSA whilst providing high degrees of skin cell viability, as well as being able to retain a high swelling capacity for exudate absorption. The inclusion of Ca^2+^ acts as a cross-linker which maintains the structural integrity of the alginate polymer, whilst acting as a source of Ca^2+^ release for fibroblast proliferation and host haemostasis. Both methods generate nanosilver particles of varying sizes and morphologies, with the smaller spherical nanosilver showing the greatest antibacterial efficacy. However, when applied to cell viability and the physiochemical characteristics of the polymer, the effects of nanoparticle size and morphology were negligible. Generally speaking, the release rates of nanosilver are lower than that of commercial products [42], but they were still able to elicit effective antibacterial activity. Based the data generated from this study, 1% *w/v* NAA samples with a stir time of less than 24 h are suggested as the most viable options for wound regeneration, as their particle size is above the penetration threshold of damaged skin, whilst providing the greatest antibacterial efficacy. This therefore suggests that the simultaneous utilisation of aloe and alginate for silver reduction is advantageous for wound regenerative applications 

Future experiments could aim to modify the current polymer matrix by increasing the nanosilver release rate and whilst adding other regenerative agents into the polymer, so as to increase its antibacterial and cellular proliferative capacity. Other suggestions include the replacement of Ca^2+^ with other cations for cross-linking, or perhaps the simultaneous utilisation of multiple cations instead [43], so as to explore the possible therapeutic release effects towards wound healing. 

## Figures and Tables

**Figure 1 pharmaceutics-15-00483-f001:**
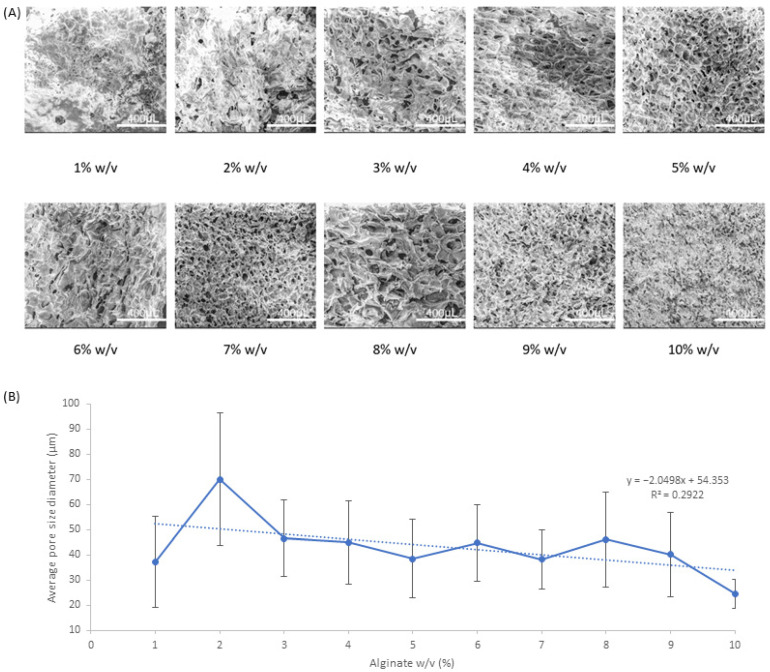
SEM analysis of NA samples (**A**) SEM images at ×200 magnification and (**B**) average pore size of samples with respect to increasing *w/v* ratio.

**Figure 2 pharmaceutics-15-00483-f002:**
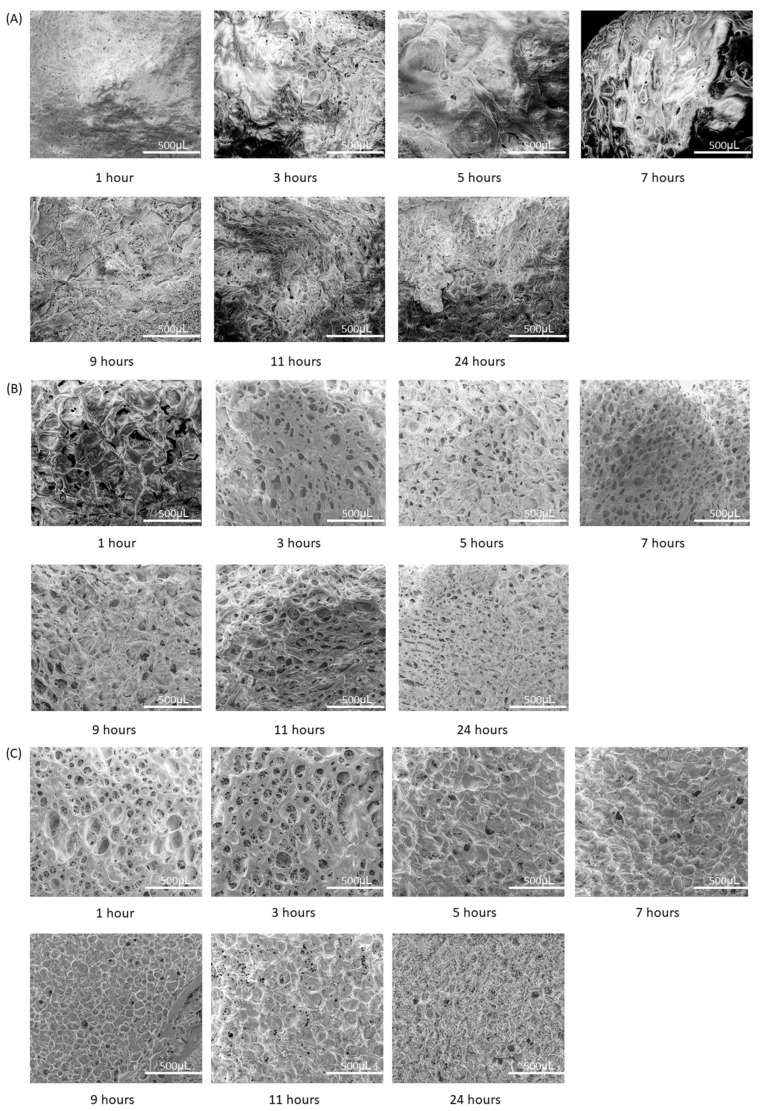
SEM analysis of NAA. (**A**) SEM images of 1% *w/v* samples at ×200 magnification, (**B**) SEM images of 5% *w/v* samples at ×200 magnification, (**C**) SEM images of 10% *w/v* samples at ×200 magnification and (**D**) average pore size of samples with respect to increasing stir time.

**Figure 3 pharmaceutics-15-00483-f003:**
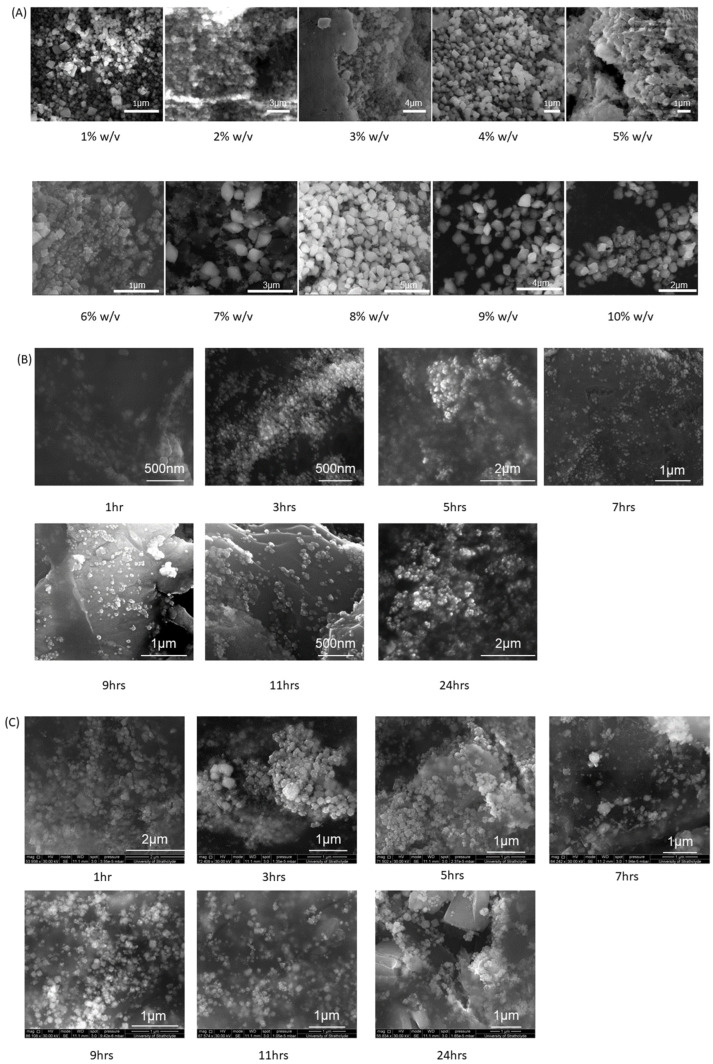
SEM images of silver nanoparticles created from: (**A**) NA 1–100%, (**B**) NAA 1% *w*/*v*, (**C**) NAA 5% *w*/*v*, and (**D**) NAA 10% *w*/*v*.

**Figure 4 pharmaceutics-15-00483-f004:**
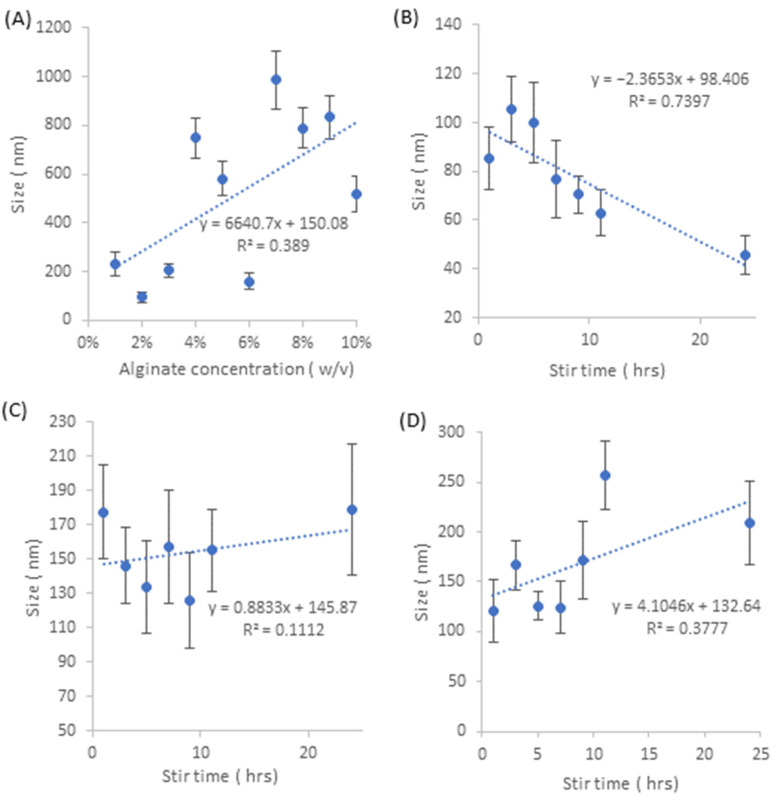
(**A**) Graph detailing the changes in particle size with respect to increasing alginate concentration. (**B**) Graph detailing the changes in particle size with respect to stirring time for NAA 1% *w*/*v*, (**C**) graph detailing the changes in particle size with respect to stirring time NAA 5% *w*/*v*, and (**D**) graph detailing the changes in particle size with respect to stirring time for NAA 10% *w*/*v*.

**Figure 5 pharmaceutics-15-00483-f005:**
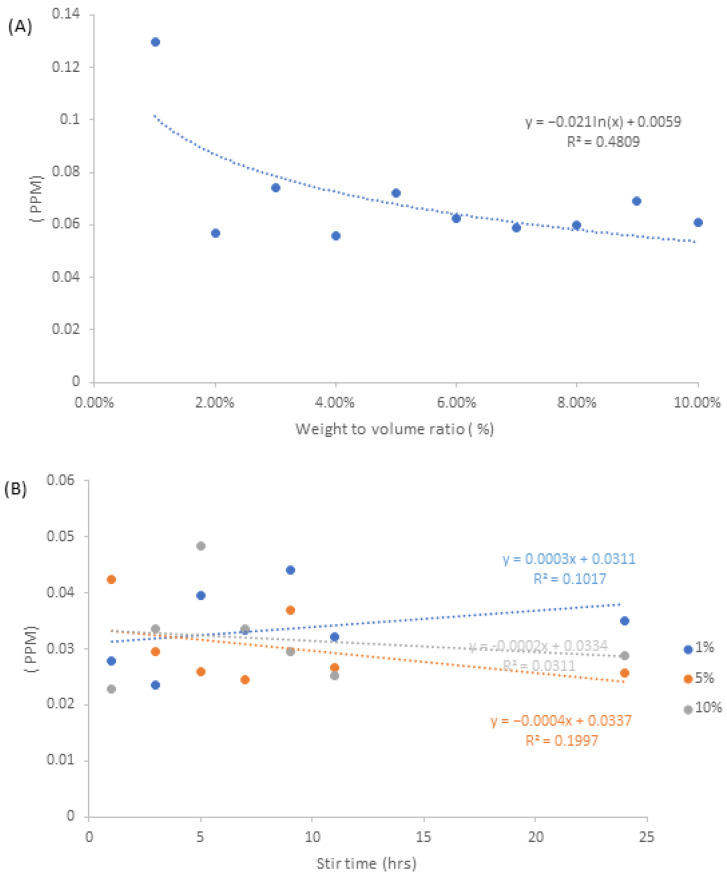
Graphs displaying the total amount of nanosilver released, which were measured at the 24 h time point for (**A**) NA samples and (**B**) NAA samples.

**Figure 6 pharmaceutics-15-00483-f006:**
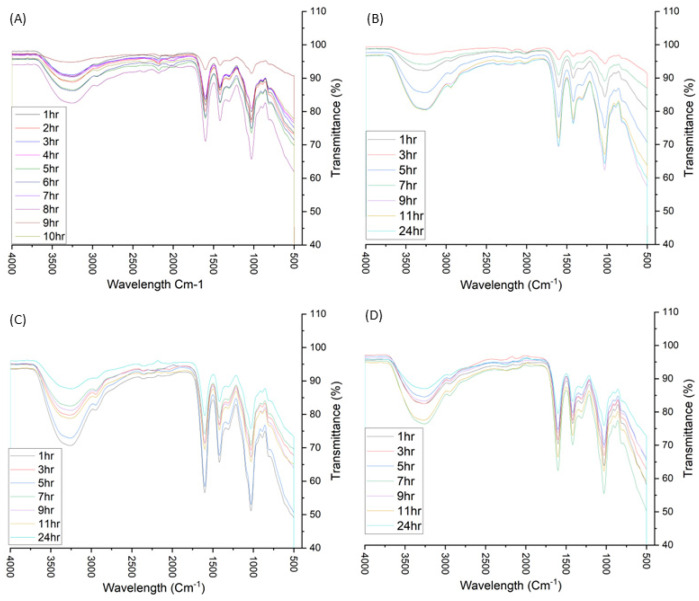
FTIR graphs for: (**A**) NA 1–10%, (**B**) 1% NAA 1–24 h, (**C**) 5% NAA 1–24 h and (**D**) 10% NAA 1–24 h.

**Figure 7 pharmaceutics-15-00483-f007:**
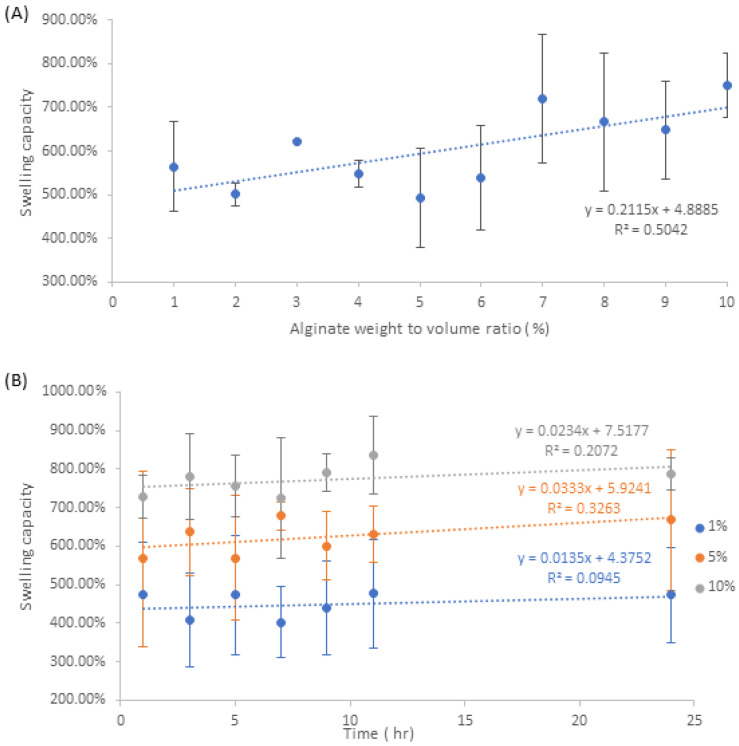
Changes in swelling capacity for (**A**) NA samples and (**B**) NAA samples.

**Figure 8 pharmaceutics-15-00483-f008:**
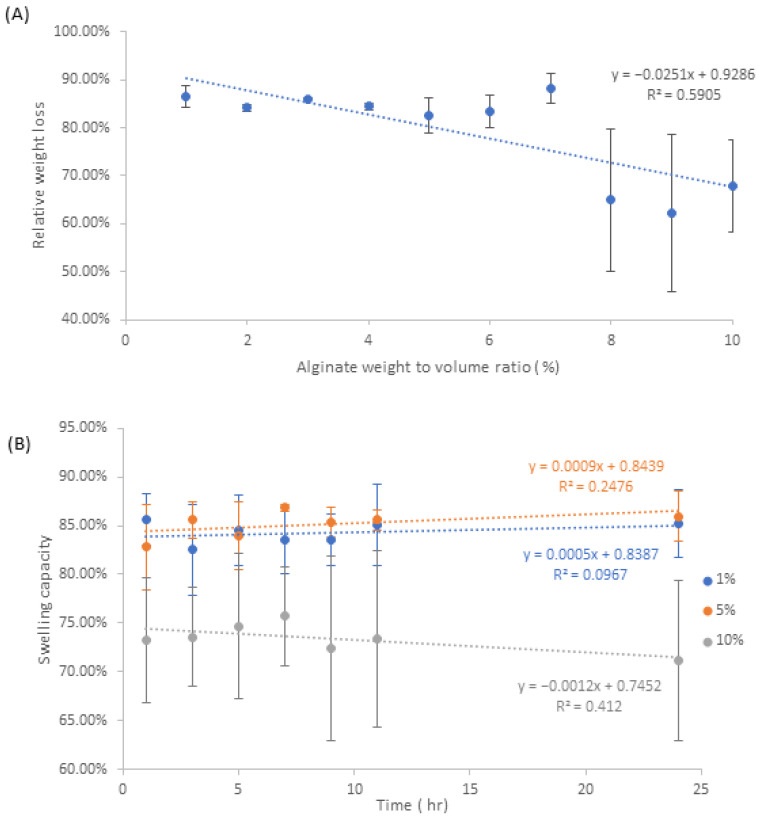
Evaporative water loss values for (**A**) NA samples and (**B**) NAA samples.

**Figure 9 pharmaceutics-15-00483-f009:**
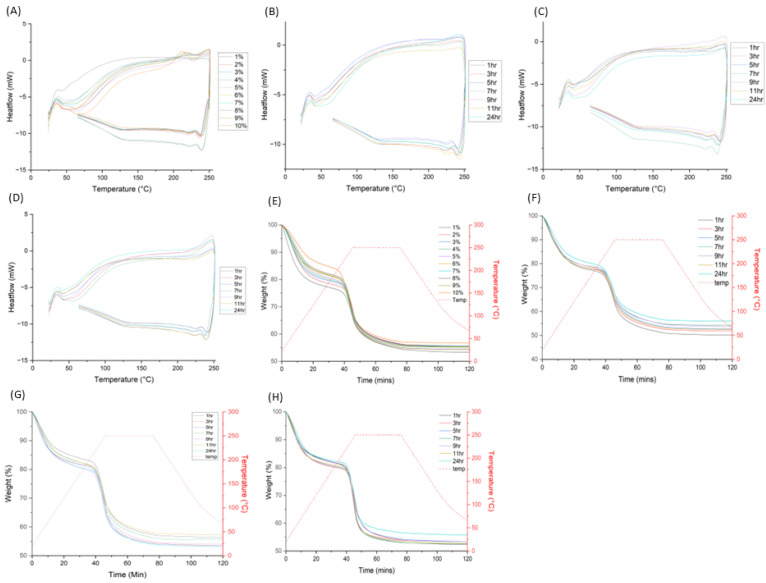
Graphs displaying the TGA and DSC analysis results for all experimental samples. (**A**) NA 1–10% DSC, (**B**) NAA 1% 1–24 h DSC, (**C**) NAA 5% 1–24 h DSC, (**D**) NAA 10% 1–24 h DSC, (**E**) NA 1–10% TGA, (**F**) NAA 1% 1–24 h TGA, (**G**) NAA 5% 1–24 h TGA and (**H**) NAA 10% 1–24 h TGA.

**Figure 10 pharmaceutics-15-00483-f010:**
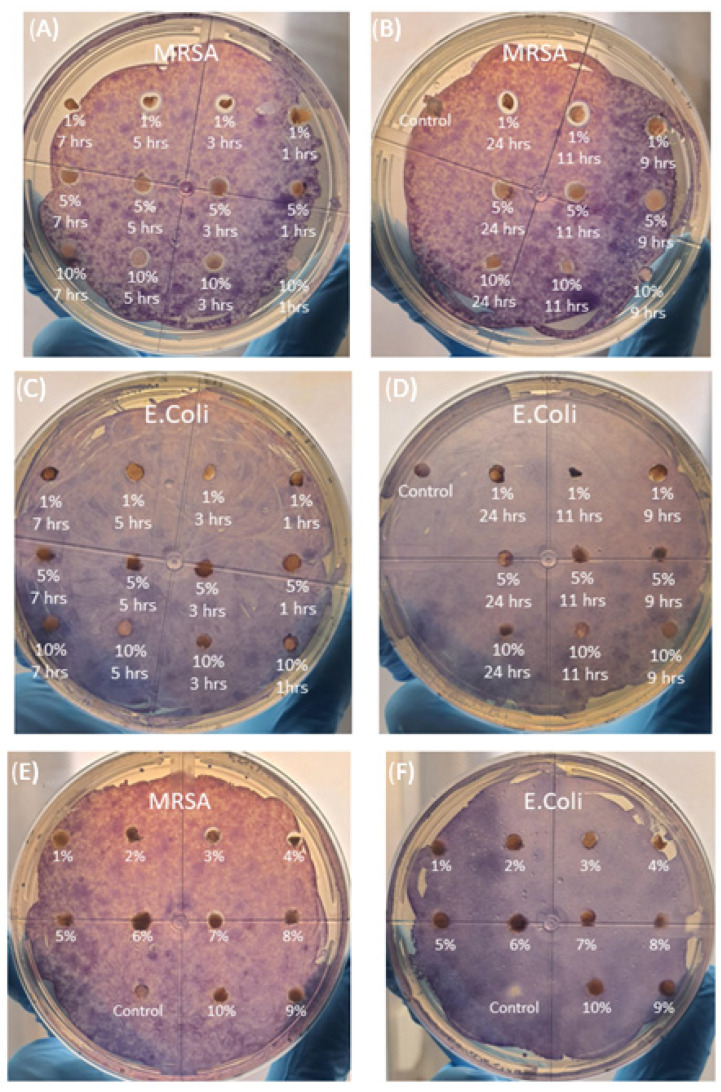
Zone of inhibition coupled with MTT. (**A**) NAA 1–7 h [MRSA], (**B**) NAA 9–24 + calcium alginate control [MRSA], (**C**) NAA% 1–7 h [*E. coli*], (**D**) NAA% 9–24 h + calcium alginate control [*E. coli*], (**E**) 1–10% NA + calcium alginate control [MRSA] and (**F**) 1–10% NA + calcium alginate control [*E. coli*].

**Figure 11 pharmaceutics-15-00483-f011:**
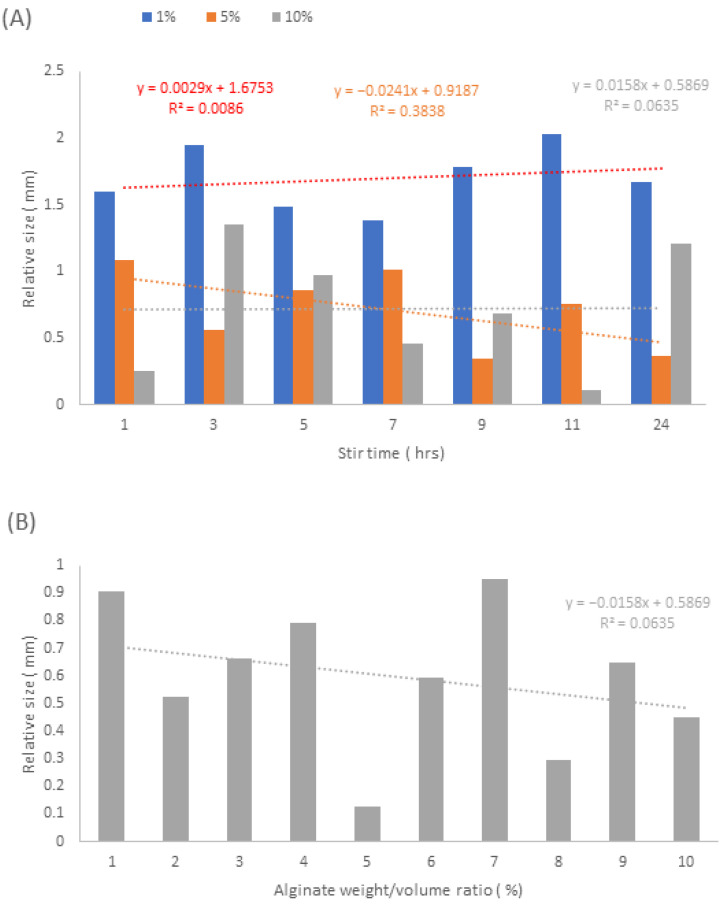
Graphs displaying the relative zone of inhibitions sizes with respect to the sample disc size. (**A**) 1%, 5% and 10% *w/v* NAA samples, (**B**) 1–10% NA samples.

**Figure 12 pharmaceutics-15-00483-f012:**
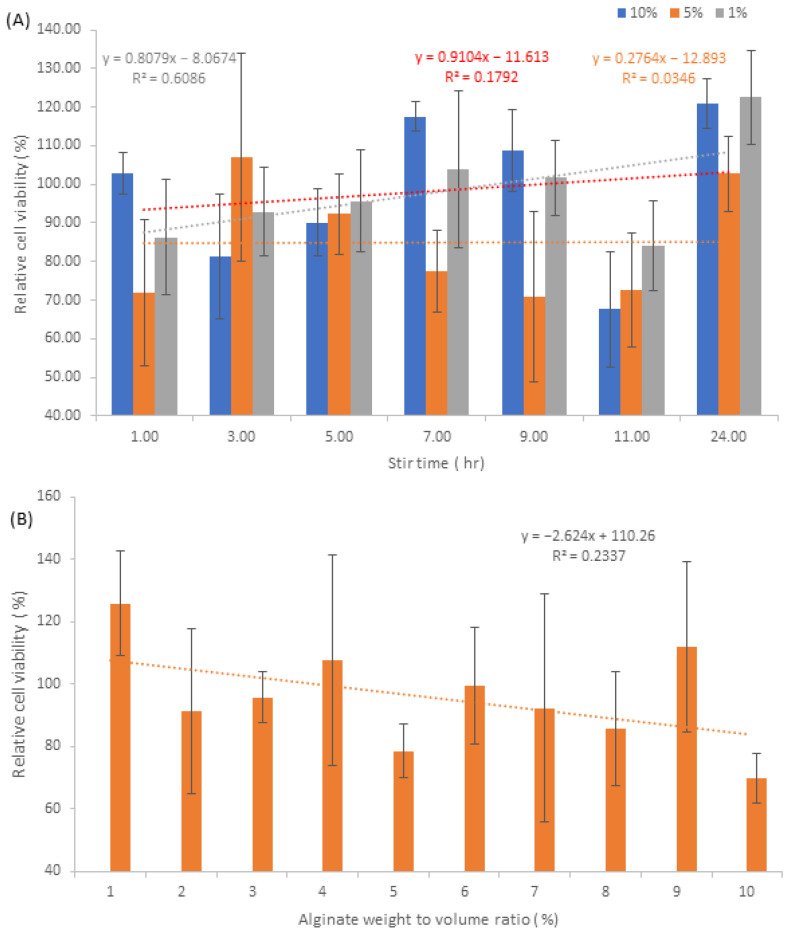
Graphs detailing average cell viability normalised against the control. (**A**) NAA samples and (**B**) NA samples.

**Figure 13 pharmaceutics-15-00483-f013:**
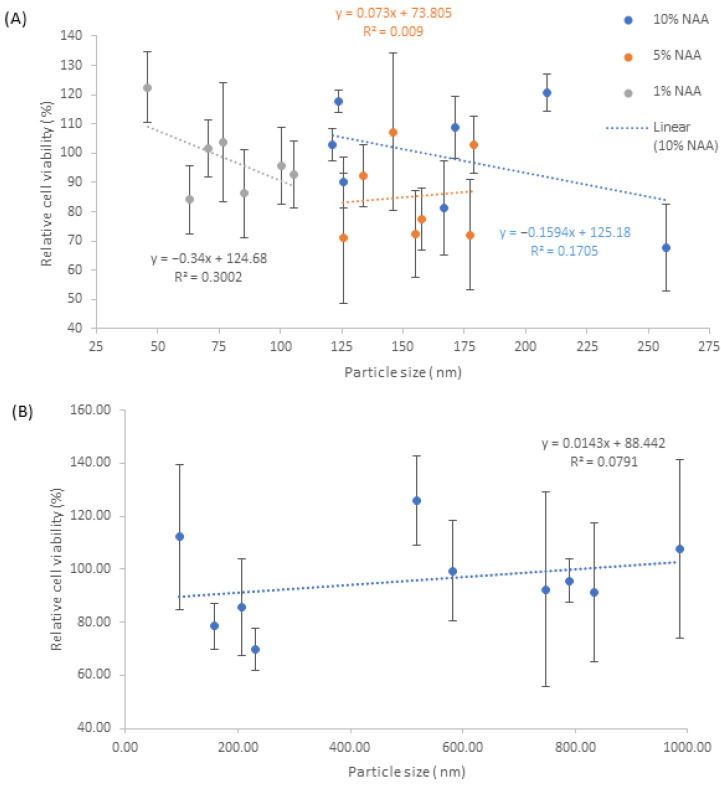
Graphs detailing average cell viability with respect to nanosilver particle size. (**A**) NAA samples and (**B**) NA samples.

**Figure 14 pharmaceutics-15-00483-f014:**
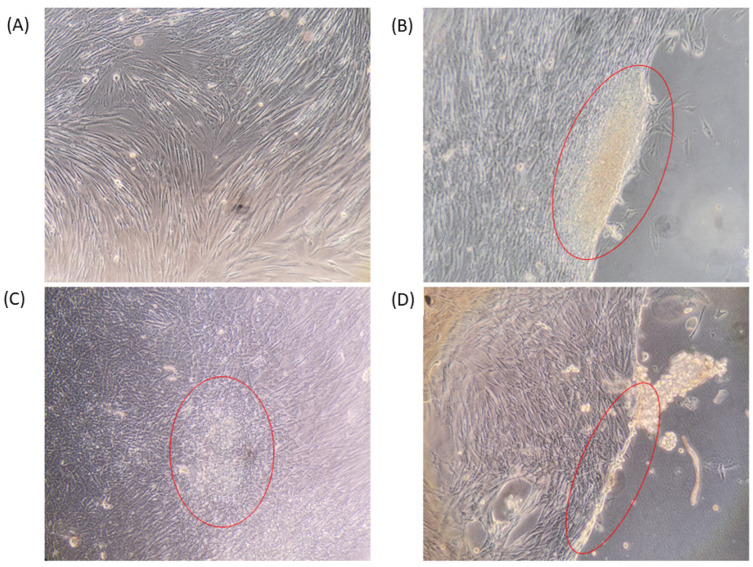
Light microscopy images at ×10 magnification detailing the HFF growth patterns. (**A**) Control with no alginate polymer and (**B**–**D**) average sample containing alginate polymer. The red rings highlight the anchor point between the cells and the polymer sample.

## Data Availability

Data is available upon request and held on the University of Strathclyde central repository.

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
