# Peer review of "Exploration of Nanosilver Calcium Alginate-Based Multifunctional Polymer Wafers for Wound Healing"

_pharmaceutics, 2023, doi:10.3390/pharmaceutics15020483_

Round 1
Reviewer 1 Report
Man et al., present a method to prepare a nanosilver calcium alginate polymer matrix and its characterization. The authors studied the effect of various compositions of nanosilver alginate matrices on their physicochemical properties, antibacterial activities against S. aureus and E. coli, and fibroblast viability. The manuscript is well-read, and the authors provide sufficient background, in-depth analyses, and thoughtful discussion.
A few questions and comments:
-The authors mentioned that there are no discernible correlations between silver release and polymer compositions. But was there any effect on the total contents of nanosilver in NA/NAA? Is it assumed that nanosilver from each composition remains similar?
-Fig5: Do nanosilver release values present in both figures show the amount of nanosilver released ppm at 24 hrs? i.e., the amount of nanosilver released was measured at 24 hr time point.
-Could the authors include the thickness of the discs used in the study?
-For the zone of inhibition studies, do images in Fig 10. show discs? Were there any other controls such as antibiotics or silver nanoparticles in solution? What is the typical antibacterial concentration of silver nanoparticles? Also, the silver release study was performed in solution, so the release could be facilitated. Would silver be readily released from the disc when placed on agar with limited exposure to water/moisture?
-Fig 12 A, x-axis title is partially blocked.
Reviewer 2 Report
1. Section 2.2 Polymer synthesis should be modified . This is not polymer synthesis, this can be formulation preparation, preparation of polymer based silver nanoparticles or nanocomposites, etc
2. Did you calculate the entrapment effeciency ?
3. How did you measure the pores, did you use SEM only or another image analysis software?
4. In section 2.3.7 you mentioned bacterial strains then 2.3.8 cell viability then 2.3.9 you returned back to inhibition zone for bacterial assessments than 2.3.10 for cell viability. This is a little bit confusing. You may preferably make a section for bacterial experiment with two subsections for bacterial strains and inhibition zone. Then make a section for cell experiments with two subsections for cell culture preparation and cell viability.
5. Please present inhibition zone and cell viability results in form of bar charts.
